# Morpho-Histology, Endogenous Hormone Dynamics and Transcriptome Profiling in Dacrydium Pectinatum during Male Cone Development

**Wenju Lu [1,†], Enbo Wang [1,†], Weijuan Zhou [1], Yifan Li [1], Zhaoji Li [1], Xiqiang Song [1], Jian Wang [1], Mingxun Ren [1], Donghua Yang [2], Shaojie Huo [1], Ying Zhao [1,\*] and Haiying Liang [3,\*]**

[1] Laboratory of Ministry of Education for Genetics and Germplasm Innovation of Tropical Special Trees and Ornamental Plants, Hainan Biological Key Laboratory for Germplasm Resources of Tropical Special Ornamental Plants, College of Forestry, Center for Terrestrial Biodiversity of the South China Sea, Hainan University, Haikou 570100, China; luwenjulu@163.com (W.L.); baroque2014@outlook.com (E.W.); zhou2677842315@163.com (W.Z.); Li_1fana@163.com (Y.L.); lizhaojino1@163.com (Z.L.); songstrong@hainanu.edu.cn (X.S.); wjhainu@hainanu.edu.cn (J.W.); renmx@hainanu.edu.cn (M.R.); huoshaojie@163.com (S.H.)

[2] Forestry Bureau of Hainan Province, Haikou 570100, China; ydh8935@126.com

[3] Department of Genetics and Biochemistry, Clemson University, Clemson, SC 29634, USA

\* Correspondence: zhaoying3732@163.com (Y.Z.); hliang@clemson.edu (H.L.)

† Wenju Lu and Enbo Wang contributed equally to this work.

**Abstract:** *Dacrydium pectinatum* de Laubenfels is a perennial gymnosperm species dominant in tropical montane rain forests. Due to severe damages by excessive deforestation, typhoons, and other external forces, the population of the species has been significantly reduced. Furthermore, its natural regeneration is poor. To better understand the male cone development in *D. pectinatum*, we examined the morphological and anatomical changes, analyzed the endogenous hormone dynamics, and profiled gene expression. The morpho-histological observations suggest that the development of *D. pectinatum* male cone can be largely divided into four stages: microspore primordium formation (April to May), microspore sac and pollen mother cell formation (July to November), pollen mother cell division (January), and pollen grain formation (February). The levels of gibberellins (GA), auxin (IAA), abscisic Acid (ABA), cytokinin (CTK), and jasmonic acid (JA) fluctuated during the process of male cone development. The first transcriptome database for a *Dacrydium* species was generated, revealing >70,000 unigene sequences. Differential expression analyses revealed several floral and hormone biosynthesis and signal transduction genes that could be critical for male cone development. Our study provides new insights on the cone development in *D. pectinatum* and the foundation for male cone induction with hormones and studies of factors contributing to the species' low rate of seed germination.

**Keywords:** conifer; gene expression; intrinsic hormone; phenology; RNA sequencing; reproductive development

## 1. Introduction

*Dacrydium pectinatum* de Laubenfels is a perennial gymnosperm in the family of Podocarpaceae. This evergreen dioecious tree is a dominant species in tropical montane rain forests and native to China (Hainan province), Malaysia (Billiton), Borneo, and Philippines (Luzon and Mindanao) [1]. An adult tree can live for more than 2000 years and grow up to 30 m tall and 3 m in diameter at breast height. The wood of *D. pectinatum* is valuable for constructing buildings and high-grade handcrafting such as ships (http://www.efloras.org/florataxon.aspx?flora_id=2&taxon_id=210000447, accessed on 1 June 2018). However, due to excessive logging and shifting cultivation, there is a signif-

icant decrease in the species' natural forest resources, and it is now listed as endangered (IUCN 3.1) by the International Union for Conservation of Nature. On the coastal plains of Sabah, Sarawak and Kalimantan, 80% of *D. pectinatum* occupancy area has lost during the massive conversion of lowland forest to oil palm plantations (https://www.iucnredlist.org/, accessed on 5 June 2018). Similarly, *D. pectinatum* natural forests have been significantly reduced in China since the 1960s due to severe damages by excessive deforestation, typhoons, and other external forces. According to a report published in 2016 (China Forestry Bureau 2016), only 14,484 hectares are available in China, accounting for 3,473,575 cubic meter of forest volume.

A survey of *D. pectinatum* found that 78% of trees in natural forests had a large diameter at breast height (>5 cm) and natural regeneration was poor in the montane rain forests [2,3]. Currently, the exact length of juvenile phase is unknown for the species, but is suspected to be long, since none of the trees with a DBH of 10 cm or smaller produced reproductive structures. Among the 180,032 *D. pectinatum* seeds collected from a natural stand in Hainan province, China, the viability and germination rate was found to be 3.11% and 0.02%, respectively [4]. These factors have seriously hindered artificial cultivation and efficient use of resources.

Efforts have been undertaken to protect *D. pectinatum* and conserve its biodiversity. However, research on this endangered species is still in its infancy. Currently, most of the published studies are focused on the activity of medicinal ingredients, seedling growth, forest community structure, genetic diversity, and origin of evolution [3,5–8]. Limited information is available on the reproduction of the species in *Dacrydium*. For example, it is unknown when reproductive buds are initiated and what factors contribute to the low seed quality and poor natural regeneration. This study aimed to help fill the vast knowledge void urgently needed for the conservation and propagation of the endangered species. We employed a combined approach of anatomy, hormone dynamics, and gene expression to study the development process of *D. pectinatum* male reproductive structure. Hormones play critical roles in development, including reproduction. For instance, applying hormones has become an effective practice to induce reproductive bud initiation, especially in species with long juvenility. Liang and Yin (1994) successfully reduced cone production age from 25 to 5 years with indole butyric acid (IBA) and zeatin ribonucleoside (ZR) in *Metasequoia glyptostroboides* [9]. Male strobili of *Cryptomeria japonica* were induced by GA$_3$ spraying onto the shoots [5]. To be effective, exogenous phytohormones need to be applied before and/or during early bud differentiation. Understanding the molecular mechanism of *D. pectinatum* cone development shall pinpoint critical genes that can be utilized in early cone induction. Because no prior large-scale *Dacrydium* genomic resources were available and RNA sequencing is a common approach of discovery of target genes, we subjected male cones at five time points to RNA sequencing, generating the first transcriptome dataset for the genus and revealing genes important in male cone development and simple-sequence repeats (SSRs) that are valuable in conservation and molecular breeding. The new insights from our study are useful in developing strategies to tackle issues such as low seed germination rate and long juvenility in *D. pectinatum*.

## 2. Materials and Methods

### 2.1. Plant Material and Growth Conditions

Male reproductive structures of *D. pectinatum* were collected monthly from early April 2018 to February 2020 at Bawangling Forest Reserve (between 18°53'~19°30' north latitude, between 108°38'~109°17' east longitude) of Changjiang County, Hainan Province, China. The collection of samples was approved by the Hainan Bawangling Nature Reserve Department, which complies with national and international standards. The species was previously confirmed by botanists and labels were present on the trees. Our conduct complied with the Convention on the Trade in Endangered Species of Wild

Fauna and Flora. Changjiang County is in a typical tropical monsoon climate zone. The annual average temperature is 24.3 °C, with 39.8 °C as the highest and 0 °C as the lowest. There is no winter throughout the year. The annual accumulated temperature is 8400~9100 °C, while the total solar radiation is 135 kcal/cm². Rainfall is abundant with an average annual precipitation of 1676 mm. Male buds were collected from north, south, east, and west sides of four mature trees that are over 100 years old based on the available records. Ten cones from each collection date were used for size measurement. Bracts were dissected and removed under a SMZ-168 stereomicroscope before buds were stored at 4 °C in 2.5% glutaraldehyde or a fixative solution (formalin: acetic acid: 70% alcohol = 1:1:18, FAA). Photos were taken of male shoots at different stages of development with a Nikon digital camera.

### 2.2. Semi-Thin Sectioning

Male buds fixed in FAA were dehydrated and embedded in paraffin, as described in [10]. Sections (~4 μm) were cut with a microtome (RM2016, Shanghai, China) and mounted on slides. After rehydration, specimens were stained with 1% saffron and 0.5% solid green and observed under a Nikon Eclipse Ci light microscope (Tokyo, Japan). Digital images were taken with a Nikon digital camera.

### 2.3. Observation of Male Cone Morphology by Scanning Electron Microscopy (SEM)

Male buds fixed in 2.5% glutaraldehyde were treated with 1% osmium acid. After dehydration with a series of increasing concentration of ethanol and dipping into iso-amyl acetate, dried samples were sputter-coated with gold prior to scanning electron microscopy examination (SEM, U8010, HITACHI, Tokyo, Japan), according to Kang et al. [11].

### 2.4. Detection of Endogenous Hormones

Male reproductive samples were collected at various time points from the four mature trees mentioned above during April 2019 to March 2020 and preserved at −80 °C. Frozen samples were ground to fine powders with a grinding machine (30 Hz, 1 min). After ground samples of the same time point were equally pooled, 50 mg were weighed and dissolved in a 0.5 mL-extract solution, containing methanol, water, and formic acid ($v$:$v$:$v$ = 15:4:1). After 10 min of extraction, supernatant was obtained by centrifugation for 5 min at 14,000 rpm. The extraction and centrifugation steps were repeated twice. All supernatants were combined and dried at 35 °C under nitrogen gas. The extracts were then resuspended with 100 μL of an 80% methanol-water solution and sonicated for 1 min, followed by filtration through a 0.22-micron polytetrafluoroethylene membrane.

IAA, CTK, ABA, and $GA_3$ were detected by enzyme-linked immunosorbent assay (ELISA), following the manufacturer's instructions (Jingmei Biotechnology, Beijing, China). JA was analyzed with an Ultra-High-Performance Liquid Chromatography (UPLC) (Shim-pack UFLC SHIMADZU CBM3OA, http//www.shimadzu.com.cn, accessed on 20 March 2019) and a Tandem Mass Spectrometry (Ms/ms) (Applied Biosystems 6500 Quadrupole Trap, http://www.applied biosystems.com.cn/, accessed on 21 March 2019). The detailed procedures are described in [12]. The hormonal contents obtained in the analysis were expressed as pg or μg per gram of fresh weight. For each time point, three samples were separately prepared and analyzed.

### 2.5. RNA Sequencing and De Novo Transcriptome Assembly

Male reproductive tissues representing the five developmental stages, microspore primordium initiation (8 April), microspore primordium development (10 May), before microspore sac formation (3 June), microspore sac formation (3 July), before pollen mother cell formation (11 November), as well as vegetative tissues (leaves) at the same time points, were harvested from three of the above-mentioned mature trees, quickly

frozen in liquid nitrogen, and then sent to the facility in Novogene (Beijing, China) for RNA extraction, cDNA library construction, and paired-end sequencing (Illumina HiSequation 4000). Samples from each tree represented one replicate. Therefore, there were three biological replicates per time point and tissue type (male bud or leaf), resulting in a total of 30 samples for RNA sequencing. Detailed protocols for RNA-seq and analyses can be found in [13]. Briefly, RN38 EASYspin Plus Plant RNA Kits (Aidlab Biotechnologies Co., Ltd., Beijing, China) were used for RNA extraction, RNase-free DNase I was applied to remove residual DNA, and the RNA integrity of these samples was assessed using the RNA Nano 6000 Assay Kit of the Agilent Bioanalyzer 2100 system (Agilent Technologies, Palo Alto, CA, USA). mRNA-Seq libraries were constructed using the NEBNextUltra RNA Library Prep Kit for Illumina (New England Biolabs, Ipswich, MA, USA).

After removal of adaptor sequences, as well as reads with >10% unknown nucleotides or bases of Q-score ≤ 10% being more than 50%, the high-quality clean data were used to perform *de novo* assembly. Transcriptome assembly was accomplished by using Trinity software with default values [14]. The resulting contigs were clustered into unigenes by Corset [15] before being compared against the following databases: National Center for Biotechnology Information (NCBI) non-redundant protein sequences (NR), NCBI non-redundant nucleotide sequences (NT), a manually annotated and reviewed protein sequence database (Swiss-Prot), Gene ontology (GO), euKaryotic Orthologous Groups (KOG), Kyoto encyclopedia of genes and genomes (KEGG), and Protein family (Pfam), with an E-value ≤ $10^{-5}$ for the functional annotation. Putative SSRs were identified by using an online tool at https://webblast.ipk-gatersleben.de/misa/ (accessed date: 5 Jane 2021). The minimum number of repeats was ten for mononucleotide repeats, six for di-nucleotide repeats, and five for tri-, tetra-, penta-, and hexa-nucleotide repeats.

### 2.6. Differentially Expressed Gene (DEG) Selection

Expression levels were calculated as fragments per kilobase of transcript per million mapped reads (FPKM) for each sample. DEGs were identified in the male reproductive tissues at five different time points, as well as in comparison to vegetative tissues. The DESeq R package (V. 1.10.1, Boston, MA, USA) was employed. Based on the negative binomial distribution model, the DESeq software estimates variance–mean dependence in count data from high-throughput sequencing and tests for differential expression [16]. The resulting *p* values were adjusted using the Benjamini and Hochberg's approach for controlling the false discovery rate. Genes with an adjusted *p*-value < 0.05 and Fold Change (FC) ≥ 1 found by DESeq were assigned as differentially expressed. All noted changes reported in the study were in the first component (A) of a comparison (A vs. B).

To verify the reliability of the RNA sequencing data, reverse transcription-quantitative polymerase chain reactions (RT-qPCR) were conducted with a set of seven MADS-box DEGs using the same tissue collections for RNA sequencing. Total RNA was extracted using a Qiagen RNeasy Mini Kit (Qiagen Inc., Valencia, CA, USA) and was reverse transcribed into cDNA by using random primers with HiScript III RT SuperMix for qPCR (+gDNA wiper) (Vazyme, Nanjing, China). The internal control was a translation elongation factor (EF1-$\alpha$). The $2^{-\Delta\Delta CT}$ method was used to analyze the data [17]. Sequences of primers are listed in Supplementary Table S1.

### 2.7. Statistical Analysis

Unless otherwise indicated, Student's *t*-test and Fisher's Least Significant Difference (LSD) multiple comparison were used for statistical analyses at the confidence level of 95%. IMB® SPSS (V. 22., Armonk, NY, USA) was utilized.

## 3. Results

### 3.1. Arrangement and Phenology of Male Cones

As described in [18], *D. pectinatum* male reproductive structures appear on the top of current-year branches. Two or three microspore bulbs cluster together, showing a V-shaped distribution, subtended by decussate bud-scales (bracts) (Figure 1A). In Bawangling Forest Reserve, male reproductive structures were first observed in early April, with a light green color and a diameter of 0.37 mm and a length of 0.32 mm. The cones gradually elongated and enlarged, and the green color deepened (Figure 1A). The cones reached their maximum length of the year in September, while the width continued to grow. In October, the outer scales of the cone became sharp. During November and until the following January, the outer scales were greatly elongated, evenly thickened, and became yellowish-brown. Furthermore, the whole structure of the male cone became more compact. Cones enlarged rapidly in spring. By February, the cones reached an average length of 8.36 mm and a diameter of 2.16 mm (Figure 1B). By late March, cones gradually expanded and became yellow-brown, with the outer scales cracked and mature microspore sacs dispersing pollens (Figure 1A). Observation of the male cone development was performed from 2018 to 2020. The SEM images showed that spirally-arranged microsporophylls were formed by April and were tightly arranged around the main axis (Figure 2). The species' pollen grains were largely round with a wrinkled surface.

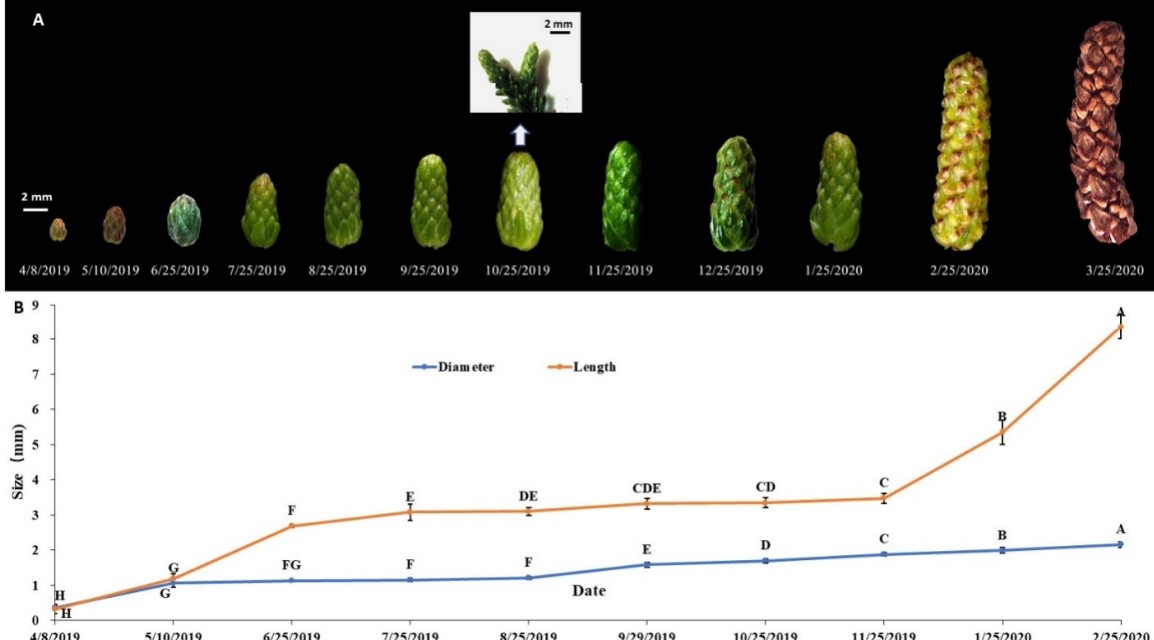

**Figure 1.** External morphological images (**A**) and size (**B**) of *D. pectinatum* male cones at different developmental stages. Specimens were collected from Changjiang County, Hainan Province, China. Different letters among length or diameter indicate significant difference at $p < 0.05$, and $n = 10$.

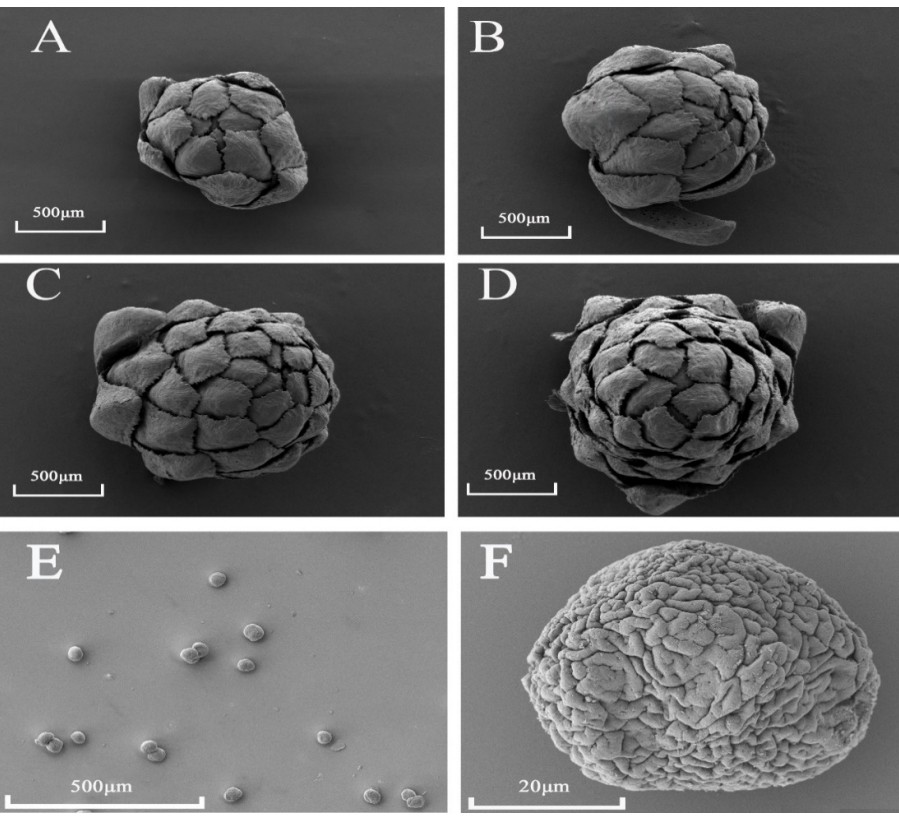

**Figure 2.** Scanning electron microscopic images of *D. pectinatum* male buds (**A**–**D**) and pollen grains (**E**,**F**). (**A**): April; (**B**): May; (**C**): July; (**D**): January; (**E**,**F**): February.

### 3.2. Microscopic Anatomy of D. pectinatum Male Cones

In April, microsporophyll primordia started to become visible, which were formed in the order of from the base of the bud to the top (Figure 3A). The bud was wrapped by phylloclades, with the left and right phylloclades along the central axis forming a U shape close to the central axis. By July, sporogenous tissues in sac-like microsporangia became condense due to the further differentiation and accelerated cell division of the microsporophyll primordia (Figure 3C–F). Microsporangia seemed to derive from a group of hypodermal cells of the microsporophyll. The formation and division of pollen mother cells were found in January of the following year. Pollen mother cells were actively dividing, producing four microspores per mother cell through meiosis (Figure 3G). Numerous pollen grains were formed in late February (Figure 3H).

The male cone had a central axis on which 15 to 20 microsporophylls were spirally arranged. Each mcrosporophyll bored one or two microsporangia on the abaxial surface. A mature microsporangium consisted of one layer of epidermis, a multilayered endothecium, tapetum, and microspore mother cells (Figure 3G,H). Tapetum separated from endothecium as microsporangia developed (Figure 3G).

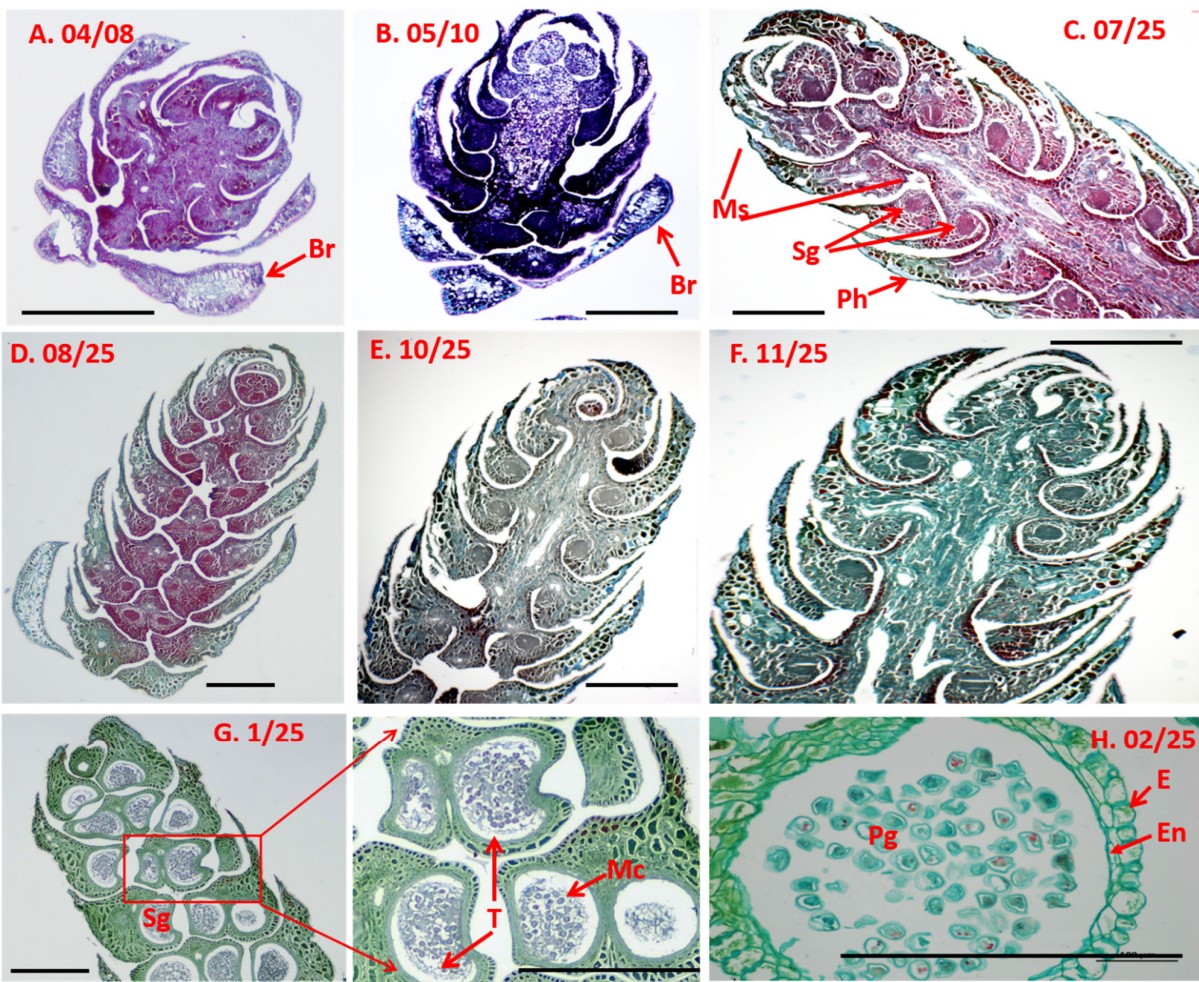

**Figure 3.** Longitudinal sectioning of *D. pectinatum* male cones at different developmental stages. (**A**,**B**): microspore primordium formation stage (April to May); (**C**–**F**): microspore sac and pollen mother cell formation stage (July to November); (**G**): pollen mother cell division; (**H**): pollen grain formation (February). Br: bract; Sg: sporangia; Ph: phylloclade; Mc: mother cell; Ms: Microspore; Pg: pollen grain; T: tapetum; En: endothecium; E: epidermis; Bars = 500 μm.

### 3.3. Dynamic of Endogenous Hormones during Male Cone Development

As male cones developed, $GA_3$ level decreased from May to June, and then increased to its peak in July. From January to March, $GA_3$ level continued to drop (Figure 4A). The IAA level decreased from April to June, and then increased. In particular, the IAA level spiked in July and January, corresponding to the microspore sac and pollen mother cell formation stage and pollen mother cell division stage, respectively (Figure 4A). CTK showed the lowest level in April. Because cone development involves active cell division, it is reasonable that CTK level was found higher at the stages after microspore primordium formation. ABA exhibited a low level in April and May during the microspore primordium formation stage. In contrast, the highest ABA level was found in November when the first year's growth paused, followed by the following year's March during pollen dispersal. There was no significant change in JA content, with an exception in late October when a large increase occurred (Figure 4B).

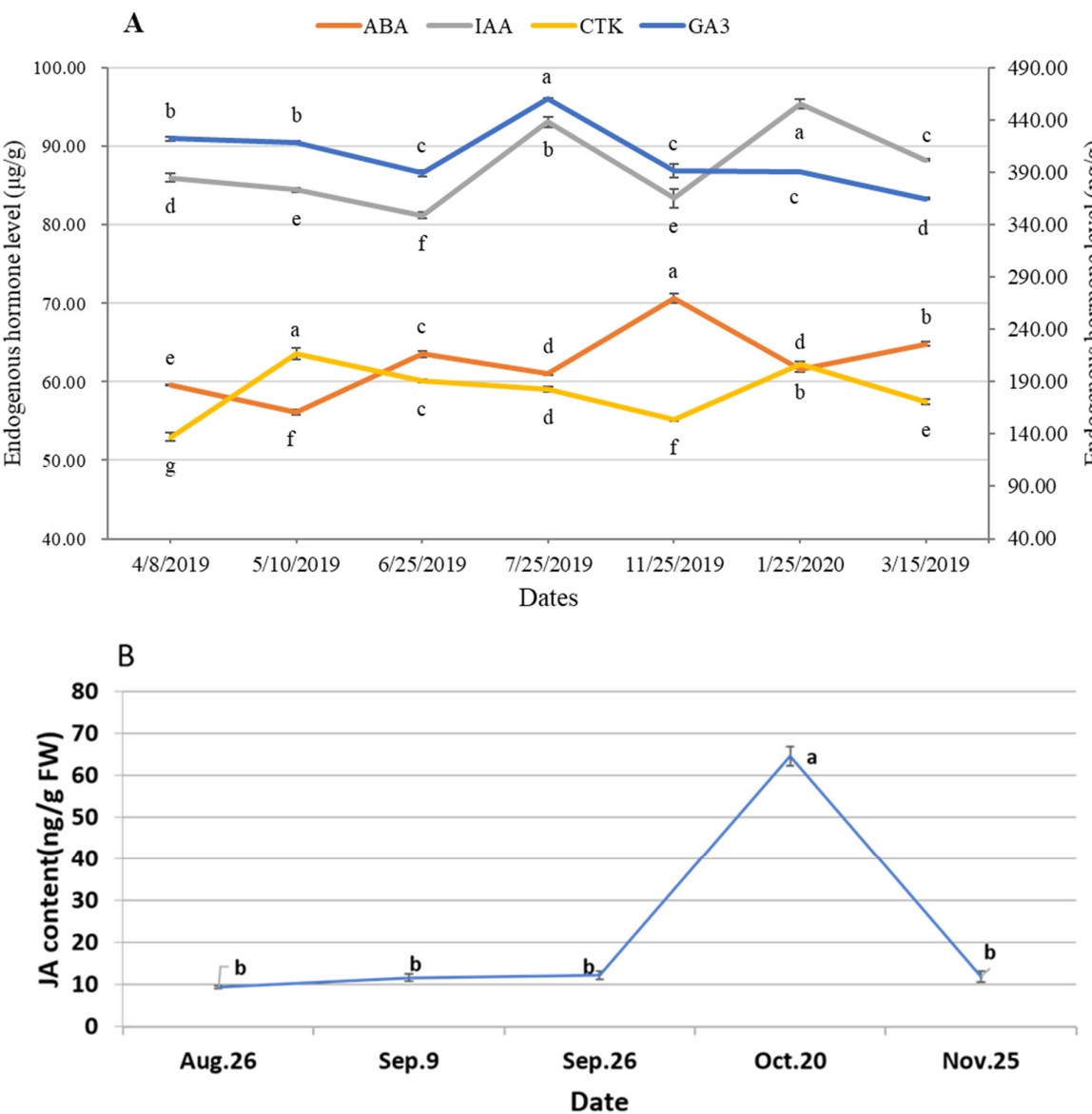

**Figure 4.** Endogenous hormone changes in *D. pectinatum* male cones at different developmental stages. (**A**) Hormones analyzed by ELISA. The unit of GA₃ was pg/g fresh weight (left vertical axis). The unit for ABA, IAA, and CTK was μg/g fresh weight (right vertical axis). (**B**) JA was detected by ultra-high-performance liquid chromatography in combination with a tandem mass spectrometry. The error bars represent standard deviation. Different letters indicate significant difference at $p < 0.05$, $n = 3$.

### 3.4. De Novo Assembly of Dacrydium Transcriptome

Sequencing of 30 samples representing leaf and male cone at five time points generated a total of 7.31 billion base pairs, with an average of 97,482,509 raw sequencing reads and 94,857,576 clean reads per sample (Supplementary Table S2). These clean reads are available in the NCBI Short Read Archive (accession number: PRJNA748147). The average ratio of clean reads to raw reads was 99.97%, and the average Q20 rate was 97.7%. The correlation among replicates was at least 0.93 (Supplementary Figure S1). In total, 70,867 unigenes were generated. The proportion of unigenes in each sample type ranged from 49.2% to 63.7% (Supplementary Figure S2). The average unigene length is

1337 bp, with 13,637 unigenes > 2000 bp, 13,071 between 1001 and 2000 bp, 20,829 between 500 and 1000 bp, and 23,330 ≤ 500 bp. The overall GC content was 44.8%.

When compared to the seven common public datasets (NR, NT, KEGG, Swiss-Prot, Pfam, GO, and KOG), 56.35% of the unigenes were annotated in at least one of the datasets, with 48.78% in NR, 40.67% each in Pfam and GO, 39.31% in Swiss-Prot, 20.55% in NT, 18.24% in KEGG, 13.83% in KOG. A set of 5003 unigenes (7.05%) found annotations in all seven datasets. Among the matches within NR, 23.8% unigenes had an E-value < $10^{-100}$, while the E-value range of $10^{-100}$ to $10^{-60}$ accounted for 16.7% of the unigenes (Figure 5A). Among the plant species surveyed in Figure 5B, the *Dacrydium* unigenes had the most matches (23.9%) with *Picea sitchensis*, a coniferous tree, followed by cork oak (*Quercus suber*) (11.6%). When the annotated functions were classified with KOG, the top four were translation, ribosomal structure and biogenesis (1757), posttranslational modification, protein turnover, and chaperones (1407), general function prediction only (1186), and energy production and conversion (929) (Supplementary Figure S3). Nuclear structure, extracellular structures, and cell mobility had the lowest unigenes, 30, 7, and 3, respectively. When annotated with KEGG, the top three pathways were ribosome (8.95%), carbon metabolism (5.00%), and biosynthesis of amino acids (3.59%) (Supplementary Table S3). In GO annotations, the top three categories in biological process were cellular process, metabolic process, and single-organism process. Cell, cell part, extracellular region part, and organelle were the top four categories in cellular component, and binding and catalytic activities were the top two in molecular function (Supplementary Figure S4). This *Dacrydium* transcriptome contains 8349 putative SSR repeats. The top four SSR categories were mononucleotide repeats with 9–12 bps (4420 SSRs), di-nucleotide repeats with 5–8 bps (1876), tri-nucleotide repeats with 5–8 bps (1325), and mononucleotide repeats with 13–16 bps (827) (Supplementary Figure S5).

There are 19,535 common unigenes among the 30 cDNA libraries included in the study, accounting for 27.6% of the total unigenes. A set of 4066 unigenes were specific to the leaf tissue, while 1300 were specific to the male reproductive tissue type. As the male cone developed, the number of time point-specific unigenes increased, with 436 in July and 519 in November samples.

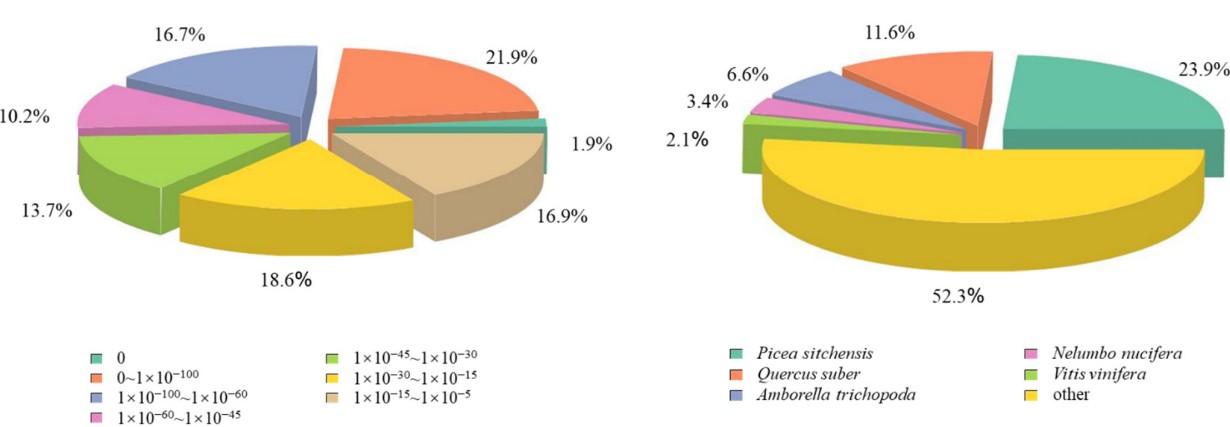

**Figure 5.** Distribution of E-values (**A**) and species classification (**B**) of *D. pectinatum* unigenes when aligned in the NCBI non-redundant protein sequences.

### 3.5. Differentially Expressed (DE) Dacrydium Unigenes

Comparisons among different timepoints and between male reproductive tissue and leaf identified a total of 74,553 DE unigenes, with 35,220 being up-regulated and

39,333 being down-regulated. As shown in Figure 6A, the comparisons of M2 vs. L2 identified the most DE unigenes, 9281. In comparison, M5 vs. M4 generated the fewest unigenes, 1615. A set of unique DE unigenes, 2535, 1738, 1467, 1431, and 859, were found in M2 vs. L2, M1 vs. L1, M3 vs. L3, M5 vs. L5, and M4 vs. L4, respectively, when comparisons were performed between male reproductive tissue and leaf at the same timepoints (Figure 6B). The number of timepoint-specific DE unigenes in male cone ranged from 381 (M5 vs. M4) to 1735 (M3 vs. M2), while in leaf the number ranged from 884 (L5 vs. L4) to 1868 (L2 vs. L1) (Figure 6C,D).

When the time points were compared sequentially within the male cone samples, a total of 10,616 unigenes were differentially expressed (Figure 6C). These DE unigenes were mapped to 117 KEGG pathways, with 10, 11, 7, and 7 pathways being significantly enriched in M2 vs. M1, M3 vs. M2, M4 vs. M3, and M5 vs. M4, respectively (Figure 7). Among these significant pathways, phenylpropanoid biosynthesis and cutin, suberin, and wax biosynthesis occurred in all four comparisons, while starch and sucrose metabolism, linoleic acid metabolism, and diterpenoid biosynthesis were found in three comparisons. When verified with RT-qPCR, 12 out of 15 comparisons had the same regulated trend as the RNA-seq data (Supplementary Table S4), indicating our RNA-seq reflected the relative differences in gene expression.

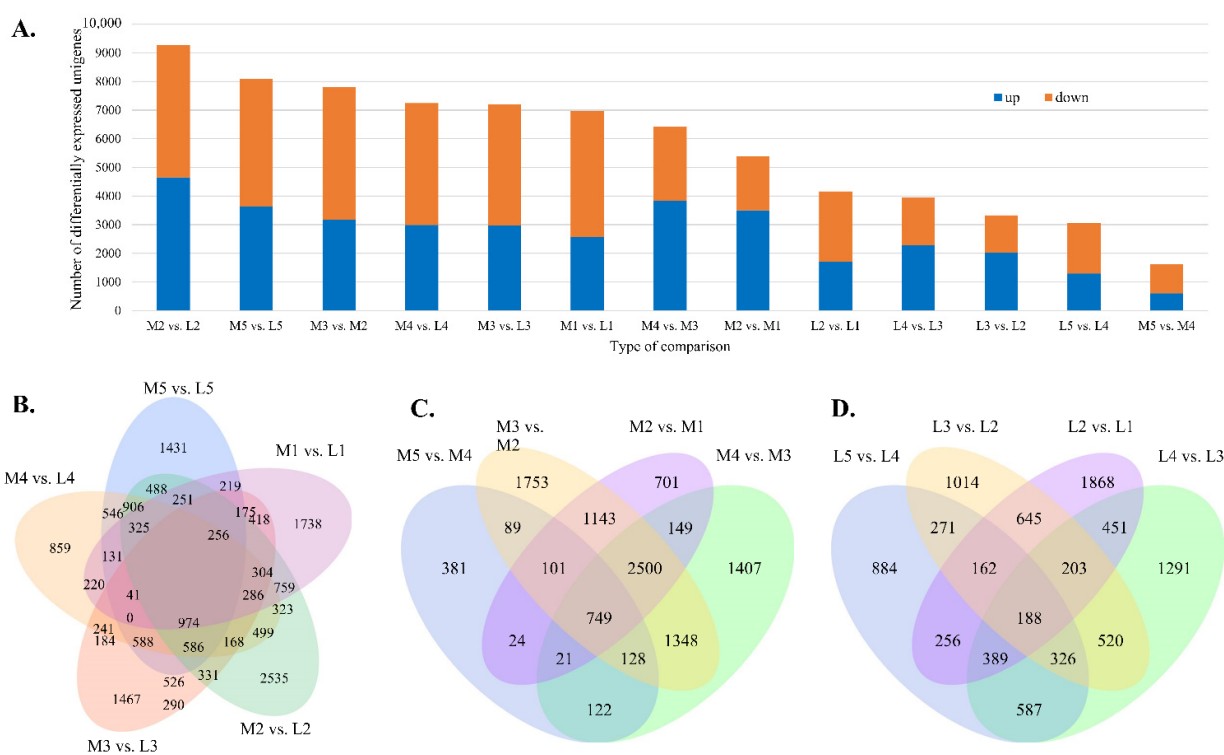

**Figure 6.** (**A**) Distribution of differentially expressed *D. pectinatum* unigenes identified in various comparisons. Venn diagram of differential genes in M vs. L (**B**), M vs. M (**C**), L vs. L (**D**). M: male reproductive tissue; L: leaf. 1: 8 April, 2: 10 May, 3: 3 June, 4: 3 July, 5: 11 November.

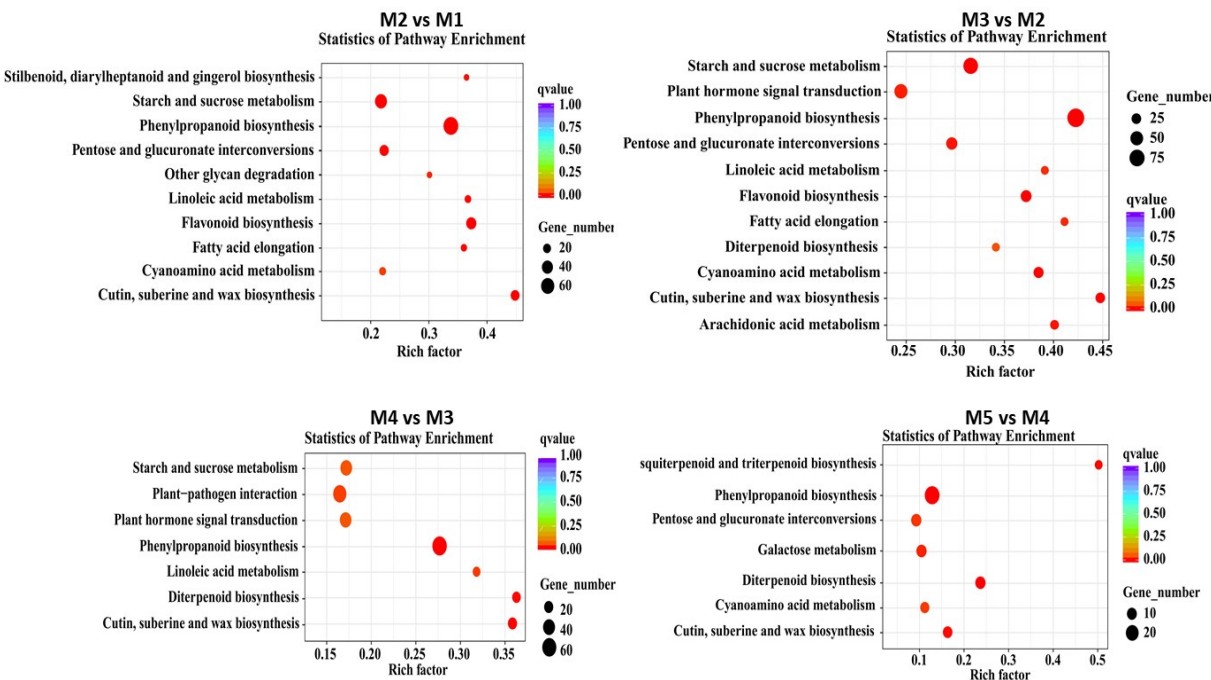

**Figure 7.** Scatter diagram of the enriched KEGG pathways in *D. pectinatum* male cones. M1: 8 April, M2: 10 May, M3: 3 June, M4: 3 July, M5: 11 November.

*3.6. DE Dacrydium Unigenes Involved in Hormone Metabolism and Hormone Signal Transduction Pathway during Male Cone Development*

Comparisons of M2 vs. M1, M3 vs. M2, M4 vs. M3, and M5 vs. M4 revealed a set of 18, 12, 32, 13, 19, and 89 DE unigenes that are involved in IAA, CTK, GA, ABA, ethylene, and jasmonic acid (JA) metabolism (Table 1). These unigenes encode for 18 different genes with 96 being up-regulated and 87 down-regulated. All these genes are in the biosynthesis pathway of endogenous hormones, except for cytokinin oxidase/dehydrogenase (CKX) gene (Supplementary Figure S6). CKX enzymes catalyze degradation of cytokinin. JA had the most DE unigenes (89), followed by GA (32).

A set of 64, 18, 13, 4, and 5 DE unigenes that are involved in IAA, CTK, GA, ABA, and ethylene signal transduction pathways, respectively, represent 1 to 4 annotated genes (Table 2). No DE unigenes were found for JA. Most of these unigenes (64) are involved in IAA signal transduction, followed by CTK (18) and GA (13). The up- and down-regulated numbers were 66 and 38, respectively. Annotations of these unigenes are listed in Supplementary Table S5.

**Table 1.** Differentially expressed *D. pectinatum* unigenes involved in hormone metabolism among male cone samples.

| Hormone Name | M2 vs. M1 | | M3 vs. M2 | | M4 vs. M3 | | M5 vs. M4 | | Total DE Unigenes | Total Annotated Genes |
|---|---|---|---|---|---|---|---|---|---|---|
| | Up | Down | Up | Down | Up | Down | Up | Down | | |
| IAA | 6 | 0 | 1 | 6 | 2 | 1 | 2 | 0 | 18 | 3 (Auxin transporter-like protein 1, Auxin transporter-like protein 2, Methylesterase 1) |
| CTK | 3 | 0 | 2 | 3 | 1 | 0 | 3 | 0 | 12 | 4 (Cytokinin trans-hydroxylase, Cytokinin riboside 5′-monophosphate phosphoribohydrolase, Cytokinin dehydrogenase, |

| | | | | | | | | | | |
|---|---|---|---|---|---|---|---|---|---|---|
| | | | | | | | | | | Adenylate kinase) |
| GA | 4 | 1 | 3 | 7 | 7 | 3 | 7 | 0 | 32 | 2 (ent-Copalyl diphosphate synthase, ent-Kaurene oxidase) |
| ABA | 1 | 5 | 3 | 2 | 1 | 1 | 0 | 0 | 13 | 4 (9-cis-epoxycarotenoid dioxygenase, Short-chain alcohol dehydrogenase/reductase, Molybdenum cofactor sulfurase, Violaxanthin de-epoxidase) |
| Ethelene | 3 | 2 | 3 | 6 | 2 | 2 | 1 | 0 | 19 | 2 (1-aminocyclopropane-1-carboxylic acid synthase, 1-aminocyclopropane-1-carboxylic acid oxidase) |
| JA | 14 | 11 | 12 | 15 | 13 | 13 | 2 | 9 | 89 | 4 (Phospholipase A1, Lipoxygenase, Allene oxide synthase, Allene oxide cyclase) |

**Table 2.** Differentially expressed *D. pectinatum* unigenes involved in hormone signal transduction pathway among male cone samples.

| Hormone Name | M2 vs. M1 | | M3 vs. M2 | | M4 vs. M3 | | M5 vs. M4 | | Total DE Unigenes | Total Annotation Categories |
|---|---|---|---|---|---|---|---|---|---|---|
| | Up | Down | Up | Down | Up | Down | Up | Down | | |
| IAA | 14 | 2 | 19 | 7 | 13 | 4 | 1 | 4 | 64 | 4 |
| CTK | 3 | 2 | 4 | 5 | 2 | 1 | 1 | 0 | 18 | 4 |
| GA | 1 | 1 | 3 | 4 | 2 | 1 | 0 | 1 | 13 | 3 |
| ABA | 0 | 0 | 2 | 1 | 0 | 1 | 0 | 0 | 4 | 1 |
| Ethylene | 0 | 2 | 1 | 0 | 0 | 1 | 0 | 1 | 5 | 2 |

*3.7. DE Unigenes Involved in the Development of Reproductive Structures*

A total of 92 flowering gene homologs were found differentially expressed within the comparisons of male samples, including *APETALA* (*AP*), *EARLY FLOWERING* (*ELF*), *LATE ELONGATED HYPOCOTYL* (*LEH*), *GIGANTEA* (*GI*), *CONSTANS* (*CO*), *FLOWERING LOCUS T* (*FT*), and *FRUITFULL* (*FUL*) (Figure 8, Supplementary Figure S7). Among the 21 DE MADS-box unigenes, *AP3* and *Agamous-like30* (*AGL30*) were highly expressed on 8 April and 3 June cones, respectively. There were 13 MADS-box unigenes with the highest expression in the May 10 samples. On each, *MADS5*, *JOINTLESS*, and *AP3-like* unigenes were exclusively expressed in cones on 11 November.

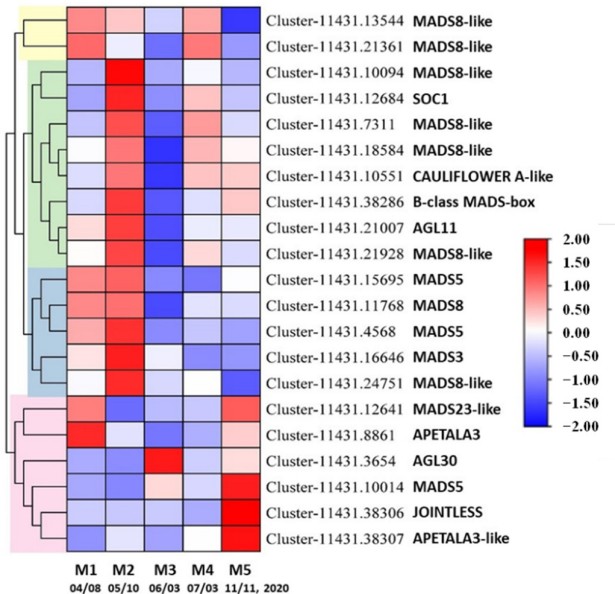

**Figure 8.** Differentially expressed *D. pectinatum* MADS-box unigenes during male cone development. SOC1: SUPPRESSOR OF OVEREXPRESSION OF CONSTANS 1; AGL: Agamous-like MADS-box protein: AP: APETALA3.

When compared to leaf samples at the same time points, a set of 18 genes in the flower development pathway were differentially expressed, with 59 up-regulations and 66 down-regulations (Figure 9). Five of these genes were found in the photoperiod pathway (light), two in the vernalization (temperature) pathway, two in the GA pathway, while six were associated with floral integrator genes and three were with floral organ identity genes. None of the genes in the nutrition pathway was differentially expressed. Notably, all *LFY* unigenes were up-regulated, and all *SOC1*, *AP2*, and *AP3* unigenes were down-regulated. Two genes in the GA pathway, gibberellic acid insensitive (*GAI*) and its repressor gene (*Repressor of GAI*), exhibited the opposite regulation.

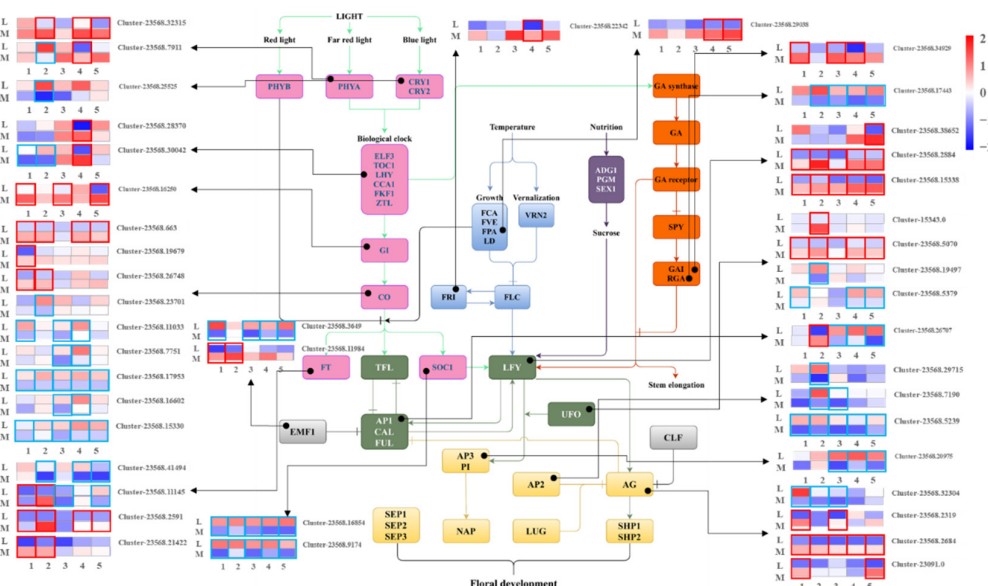

**Figure 9.** Differentially expressed flowering *D. pectinatum* unigenes identified in comparisons between leaf and male cone tissues. The gradient-colored barcode at the top right indicates the log2(FC) value. Fold change is calculated based on the difference multiple of FPKM in different periods. The unigenes expression in male cone was significantly higher

than leaves, which was framed in red box, and lower was in blue box. 1~5: 8 April, 10 May, 3 June, 3 July, and 11 November 2020; M: male; L; leaves. PHYA/B: Phytochrome A/B; CRY1/2: Cryptochrome 1/2; ELF3: Early flowering 3; TOC1: Timing of CAB1; LHY: Late elongated hypocotyl; CCA1: Circadian clock-associated 1; FKF1: Flavin-binding kelch repeat F-box1; ZTL: Zeitlupe; GI: Gigantea; CO: Constans; FT: Flowering locus T; SOC1: Suppressor of constans overexpression 1; FCA: Flowering control local A; FVE: MSI1-homologue, (a conserved WD-repeat protein found in many chromatin complexes); FPA: RRM-domain proteins; LD: Luminidependens; VRN2: Vernalization 2; FLC: Flowering locus C; FRI: Frigida; ADG1: ADP glucose pyrophosphorylase; PGM: Phosphoglucomutase; SEX1: Starch excess 1; GA: gibberellin; SPY: Spindly; GAI: Gibberellic acid insensitive; RGA: Repressor of GA; TFL: Terminal flower 1; LFY: Leafy; AP1/2/3: Apetala 1/2/3; CAL: Cauliflower; FUL: Fruitfull; UFO: Unusual floral organs; EMF1: Embryonic flower 1; CLF: Curly leaf; PI: Pistillata; AG: Agamous; SEP1/2/3: Sepallata1/2/3; NAP: NAC-like activated by AP3/PI; LUG: Leunig; SHP1/2: Shatterproof 1/2.

## 4. Discussion

### 4.1. Development of D. pectinatum Male Cone Lasts for a Year and Its Microsporophylls Are Spirally Arranged

There are 21 species in the genus *Dacrydium* (https://www.conifers.org/po/Dacrydium.php) accessed on 4 Jane 2020. Their natural distribution ranges from New Zealand, New Caledonia, Fiji and the Solomon Islands through New Guinea, Indonesia, Malaysia, and the Philippines, to Thailand and southern China [19]. Currently, not much information is available about when the initiation occurs for reproductive cones in a *Dacrydium* species. In New Zealand, *D. cupressinum* cone initiation is suggested to occur in late summer or autumn with pollination occurring in spring [20]. Our morpho-histological study indicates that *D. pectinatum* male cones initiate before April in Hainan Island, China, because male buds are distinguishable by early April (Figure 1A). Different species and climates can be the contributing factors for the discrepancy observed. Buds will be collected in February and March in the future for examination so that the exact initiation period for *D. pectinatum* male strobili can be determined. Mature *D. pectinatum* male cones average 8.5 mm in length (Figure 1B), similar to de Laubenfels's report for the same species (6–12 mm) [18], while longer than *D. Bidwillii*'s (2 to 6 mm). When more information becomes available for other *Dacrydium* species, it will be interesting to compare their reproductive buds and cones' morphology and phenology. This will help understand the diversity among the genus.

Similar to other coniferous species such as *Metasequoia glyptostroboides*, *D. pectinatum* has microsporophylls spirally arranged around a main axis, and each microsporophyll consists of a phylloclade at the apex and one or two microsporangia at the base (Figures 2 and 3). Like *M. glyptostroboides* [21], *D. pectinatum* male cones are mainly located around the outer and sunlit parts of crown. This is advantageous for pollen dispersal by wind, which is common in conifers [22,23]. Our study suggests that the development of *D. pectinatum* microspore can be divided into four stages: initiation and differentiation of microsporophyll primordia, microspore sac and pollen mother cell formation, division of pollen mother cells, and pollen grain formation. This process lasts for about 12 months, and it seems male cone initiation and pollen dispersal overlap during March and April. The long development period can arguably be a contributing factor to the low seed quality of the species, considering that many adverse factors such extreme weathers and diseases can occur, affecting the quality and quantity of pollen production. We will conduct a similar study on female buds and cones in the near future. The new information will provide further insight into the seed development of the species. A phenological comparison between male and female cones will shed light on the overlapping period of pollen dispersal and female cone opening and its conceivable role in the species' poor seed quality.

### 4.2. Endogenous Hormones Fluctuate during the Process of Male Cone Development

Plant hormones and their interplay have important roles in various aspects of plant growth, development, and reproductive processes. Major phytohormones include auxins, abscisic acid, cytokinin, ethylene, gibberellin, brassinosteroids, jasmonates, and strigolactones [24,25]. Effects of hormones and their balance on reproductive bud initiation and development seem to depend on species and sex. In female *Gnetum parvifolium* buds, $GA_3$ and ABA levels decline, and IAA increases as development progresses. In contrast, these endogenous hormones have the opposite trends in male buds [26]. High levels of $GA_3$ are reported to be beneficial for male cone formation in *G. parvifolium* [26] and other conifers, such as *Pinus* [27], Douglas-fir [28], and white spruce [29]. In our previous study with *Metasequoia*, higher levels of $GA_{1+3}$ and lower levels of IAA and ABA were beneficial to male primordium initiation, while higher levels of IAA and $GA_{1+3}$ and a lower level of ABA were favorable to female cone initiation [9]. In *D. pectinatum*, level of GA, IAA, ABA, CTK, and JA fluctuated during the process of male cone development, suggesting different development stages require various dosages and interplay of endogenous hormones. ABA seems to have an antagonistic relationship with $GA_3$, IAA, and CTK, because the ABA dynamics was largely the opposite of the others (Figure 4A). Our data corroborate the findings in lodgepole pine [30].

It is noteworthy that there was a dramatic increase of JA in male buds collected in late October during the pollen mother cell formation stage (Figure 4B). JA is well known for its roles in a plant's biotic responses, such as drought, salt stress, low temperatures [31,32]. More recently, there is increasing evidence indicating JA's involvement in plant development and reproduction. According to the reviews by Huang et al. [33] and Yuan and Zhang [34], JA is found in control of stamen development and inhibition of petal expansion in *Arabidopsis*, sex determination in maize, and control of stamen and spikelet development in rice, as well as regulates embryo/seed development and induction of leaf senescence. The actions of JA can be channeled through its signaling pathway. To our knowledge, no reports of similar information are available for a gymnosperm species. Therefore, the JA surge in *D. pectinatum* shall be further investigated.

### 4.3. RNA Sequencing Reveals Many Dacrydium Genes and SSR Makers

A search of "Dacrydium" in the NCBI databases resulted in only 303 nucleotide sequences as June 2021. Almost all these sequences either belong to chloroplast genome or are microsatellite sequences. Therefore, nearly all of the >70,000 unigenes generated from our RNA sequencing of 30 samples were revealed for the first time, greatly enhancing the genomic resource for the genus. A total of 19,535 unigenes were found common among all the cDNA libraries. A set of 4066 unigenes were specific to the leaf tissue, and there were 1300 specific to the male reproductive tissue type. To our knowledge, this is the only *Dacrydium* transcriptome being reported. When better characterized, the unigenes can be applied in understanding the molecular mechanisms of various important biological processes in *Dacrydium*. Prior to our study, only 15 *D. pectinatum* SSRs have been developed [5]. When validated, the >8000 SSRs identified in the transcriptome can be applied in the studies of biodiversity and conservation of this endangered species, as well as to accelerate selection in marker-assisted breeding. For example, informative SSRs can be utilized in determining how far pollens and seeds are dispersed in a stand. This question is relevant to natural forest regeneration, especially for dioecious species such as *D. pectinatum*. The initial glimpse into the complexity of the *D. pectinatum* genome, we found that the GC content of the *D. pectinatum* transcriptome was 45%, similar to what have been reported in other conifer species, e.g., 47% in *Amentotaxus argotaenia* [35] and 44.58% in *Pinus dabeshanensis* [36]. The dominant SSR type was mononucleotide repeats with 9–12 bps.

*4.4. Gene Expression Is Modulated during D. pectinatum Male Cone Development*

A total of 188,906 DE unigenes were found between leaf and male cone tissues (Figure 6A). Because a separate study is undertaken to compare among female and male cones and leaf in term of morphology, histology, hormone dynamics, and gene expression, we focused on the development of male cone in this current study. The M3 (June) vs. M2 (May) comparison generated the most DE unigenes. These two time points reflect the transition from microspore primordium formation stage to microspore sac and pollen mother cell formation stage (Figure 3). The fact that hormone signal transduction was found significantly enriched in the M3 vs. M2 comparison (Figure 7) underscores the importance of hormones in this transition. When genes in the metabolic pathways of hormones were examined, 17 out of 18 *D. Pectinatum* DE unigenes were found in the biosynthesis process. This suggests that the observed fluctuations in hormone levels during male cone development are mainly due to changes in their synthesis. Because 16 of the 18 DE hormone metabolism genes contains multiple unigenes with different expression patterns (Supplementary Figure S6), further studies are warranted to better understand their roles. Considering that CTK maintained a higher level during male cone development when compared to the earliest timepoint (April) and had the most DE annotated genes in both hormone metabolism and signal transduction, CTK may be effective in male cone induction in juvenile *D. Pectinatum* trees. After all, CTK has been applied in *Metasequoia* [9] and *Pinus* [37,38] for shortening juvenility.

Among the 92 DE flowering gene homologs identified among comparisons of male cone samples (Figure 8, Supplementary Figure S7), 36 are in the photoperiod pathway, 21 in hormone GA pathway, and 12 in autonomous pathway, suggesting the importance of these pathways in the male cone initiation and development. Several DE MADS-box homologs of the timepoint-specific unigenes were found to be exclusively expressed in the male reproductive part in other species, including *AP3* in *Taxus chinensis* [39] and *AGL30* in *Arabidopsis* [40]. *AP3* orthologs in *Pinus tabuliformis* and *Picea abies*, *Deficiens-Agamous-Like11* (*DAL11*), and *DAL13* are also male specific [41,42]. Both studies by Fei et al. [35] and Verelst et al. [36] found the expression of *AP3* and *AGL30* reaching its peak during the meiosis of microsporocyte. In comparison, the *D. pectinatum AP3* and *AGL30* homologs were found expressed in an earlier stage (Figure 8). MADS5 (AGL15) is known to promote somatic embryogenesis and negatively regulates auxin signaling in *Arabidopsis* and soybean [43]. Interestingly, we observed a significant decrease of IAA on 15 November following the peak expression of *MADS5* on 11 November. *JOINTLESS* is a MADS-box gene controlling flower abscission zone development [44,45]. Thus, it is reasonable to see its expression in the 11 November samples when male cones started to reach the end of its development. In March, male cones start to drop from trees after pollen dispersal. The eight *MADS8/MADS8-like D. pectinatum* homologs were largely highly expressed in the 8 April and/or 10 May samples, suggesting their roles in the microspore primordium formation. The function of *MADS8/MADS8-like* genes remains to be demonstrated. In apple, it was found that suppression of *MADS8* and *MADS9* led to sepaloid petals and greatly reduced fruit flesh [46].

Comparisons of male cone samples with leaves at the same time points also identified photoperiod pathway having the most DE unigenes (Figure 9), further suggesting its importance in the male cone initiation and development of *D. pectinatum*. Because no gene in the nutrition pathway was differentially expressed, it is speculated that its role is limited. In consistence with *LFY*'s essential roles in floral meristem identify and organ identify, two of the three *D. pectinatum LFY* unigenes were up-regulated across the five times points being surveyed. Considering that GAI is a repressor of GA responses and inhibits the expression of *LFY*, the results of *GAI* being up-regulated in three times points and its repressor (*RGA*) being down-regulated in four times points imply that GA inhibitors such as paclobutrazol (PBZ) can be applied to induce early cone formation in *D. pectinatum*. After all, PBZ has been found effective in early flowering in several woody species such as camellias, apple, and mango [47–51].

## 5. Conclusions

The lack of knowledge on *D. pectinatum* reproduction has become a challenge in conservation and propagation of the endangered gymnosperm. In this study, we report that male buds become distinguishable in April in the tropical montane rain forests in China and continue to differentiate and develop until the following March. The dynamic change of endogenous hormones suggests their roles in different stages of cone development. Cone induction with hormones may provide an alternate approach to address the seed shortage issue due to the species' long juvenility. It is suggested that treatments, maybe with CTK or PBZ, for male cone induction should be applied no later than April, before the differentiation of vegetative and reproductive buds. The RNA sequencing analyses generated the first transcriptome database for *Dacrydium* and revealed several floral and hormone biosynthesis and signal transduction genes that could be critical for male cone development. The large-scale genomic resource will enable opportunities for genetic and genomic studies of the genus. The floral genes are valuable not only to the understanding and manipulation of *Dacrydium* cone formation, but also to the evolutionary process of flowering development. Our study is of significance, representing a starting point for in-depth analyses of the species' reproductive development that will help tackle the challenges of low seed quality and poor natural regeneration.

**Supplementary Materials:** The following are available online at www.mdpi.com/article/10.3390/f12111598/s1. Figure S1: Pearson correlation of the *Dacrydium pectinatum* RNAseq samples. Figure S2: Percentage of unigenes in each *Dacrydium pectinatum* sample type. Figure S3: Function classification of *D. pectinatum* unigenes with euKaryotic Orthologous Groups (KOG). Figure S4: GO annotation of *Dacrydium pectinatum* unigenes. Figure S5: Distribution of simple sequence repeats (SSRs) identified in the transcriptome of *Dacrydium pectinatum*. Figure S6: Expression dynamics of *D. pectinatum* floral gene homologs differentially expressed during male cone development. Figure S7: Differentially expressed *Dacrydium pectinatum* unigenes identified in hormone metabolic pathways. Table S1: Sequences of primers used for RT-qPCR. Table S2: Summary of RNA sequencing data of *Dacrydium pectinatum* leave and male cone tissues. Table S3: KEGG pathway annotation of *Dacrydium pectinatum* unigenes. Table S4: RT-qPCR results of seven MADS-box unigenes and their comparisons with RNA-seq analysis. Table S5: Differentially expressed unigenes during *D. pectinatum* male cone development that are involved in plant hormones biosynthesis or metabolism.

**Author Contributions:** Conceptualization, W.L. and E.W.; methodology, W.Z., Y.L., and Z.L.; software, D.Y. and S.H.; formal analysis, M.R. and E.W.; validation and investigation, X.S. and J.W.; writing, Y.Z. and H.L. All authors have read and agreed to the published version of the manuscript.

**Funding:** This study was jointly supported by National Natural Science Foundation of China (31760217), Hainan province key research and development plan (ZDYF2020099 and ZDYF2020152), National Natural Science Foundation of Hainan Province (320RC471 and 2019RC004), Central Public-interest Scientific Institution Basal Research Fund for Chinese Academy of Tropical Agricultural Sciences (1630052021015), Innovative Research Team Program of Hainan Natural Science Fund (2018CXTD331). The supporters played an important role in the design, analysis, or interpretation of this study and the relevant data.

**Institutional Review Board Statement:** Not applicable.

**Informed Consent Statement:** Not applicable.

**Data Availability Statement:** Male reproductive samples at various developmental stages in Dacrydium pectinatum de Laubenfels. BioProject: PRJNA748147 (https://dataview.ncbi.nlm.nih.gov/object/PRJNA748147?reviewer=nkvg600rtbs8ao5qpobeklfv5s) accessed on 19 November 2021.

**Conflicts of Interest:** The authors declare no conflict of interest.

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
