# Peer review of "Morpho-Histology, Endogenous Hormone Dynamics, and Transcriptome Profiling in Dacrydium Pectinatum during Male Cone Development"

_forests, doi:10.3390/f12111598_

Round 1
Reviewer 1 Report
I've recognized that the authors provided the background information of the study and discussed the usefulness of the study for the conservation of the species. I now can understand the motivation of the study.
I have some concerns in the current version of manuscript as follows:
L. 126 and L. 146 - It is not clear which individual trees are used for hormone measurements and RNA-seq. Did you collect tissue samples for RNA-seq from an identical individual throughout the experiment? Did you use the same tree individual for hormone measurements and RNA-seq?
L. 141 - "The detailed procedures described in [12] were followed." -> The detailed procedures are described in [12].
L. 143 and 153 "three biological replicates" - Do you mean 'three leaves from a tree' or 'one leaf from each of three plants' by "three biological replicates"?
L. 153 "tissue type" - Please mention that leaves and male buds are collected.
L. 165 - Did you count the number of reads per isotig? If so, please replace "unigenes" with "isoforms" or "isotigs" throughout the manuscript to avoid confusion.
L. 303 - Please mention the aims and methods of SSR identification in the Introduction and Methods.
L. 341 "indicating the DEGs obtained were reliable" -> indicating our RNA-seq reflected the relative differences in gene expression.
Fig 6 - This figure should be moved to Supplemental information if the authors do not discuss the characteristics of the whole transcripts.
Fig 7A — Please explain why you need to compare gene expression between different tissues across developmental stages (e.g., M2 vs L1, M1 vs. L3). I think only the comparisons between leaf and reproductive tissues within a same developmental stage (e.g., M1 vs. L1) or those between the sample tissue types (e.g., L2 vs. L1, M2 vs. M1) are useful.
Fig 7A and text - Please clarify which of two members in a comparison corresponds to "up" or "down".
Tables 1-2 and Figs 8-10 - How many genes that expressed in L1 and L2 are included in the 701 DEGs between M2 and M1. What about for the other three comparisons? I suggest the authors should remove those genes from the pathway analyses.
Author Response
Comment 1---L. 126 and L. 146 - It is not clear which individual trees are used for hormone measurements and RNA-seq. Did you collect tissue samples for RNA-seq from an identical individual throughout the experiment? Did you use the same tree individual for hormone measurements and RNA-seq?
Response 1: we have provided this missing information in the revised manuscript.
Comment 2---L. 141 - "The detailed procedures described in [12] were followed." -> The detailed procedures are described in [12].
Response 2: We have replaced this sentence.
Comment 3---L. 143 and 153 "three biological replicates" - Do you mean 'three leaves from a tree' or 'one leaf from each of three plants' by "three biological replicates"?
Response 3: we have provided this missing information in the revised manuscript.
Comment 4---L. 153 "tissue type" - Please mention that leaves and male buds are collected.
Response 4: “tissue type” now is replaced with “issue type (male bud or leaf)” (L153).
Comment 5---L. 165 - Did you count the number of reads per isotig? If so, please replace "unigenes" with "isoforms" or "isotigs" throughout the manuscript to avoid confusion.
Response 5: In the Method, we have added that the contigs derived from Trinity were clustered by Corset, a software that generated unigenes for the downstream analyses.
Comment 6---L. 303 - Please mention the aims and methods of SSR identification in the Introduction and Methods.
Response 6: This missing information has been added.
Comment 7---L. 341 "indicating the DEGs obtained were reliable" -> indicating our RNA-seq reflected the relative differences in gene expression.
Response 7: We have revised this sentence.
Comment 8---Fig 6 - This figure should be moved to Supplemental information if the authors do not discuss the characteristics of the whole transcripts.
Response 8: this suggestion has been incorporated in the revised manuscript.
Comment 9---Fig 7A — Please explain why you need to compare gene expression between different tissues across developmental stages (e.g., M2 vs L1, M1 vs. L3). I think only the comparisons between leaf and reproductive tissues within a same developmental stage (e.g., M1 vs. L1) or those between the sample tissue types (e.g., L2 vs. L1, M2 vs. M1) are useful.
Response 9: we included all the possible comparisons in Fig. 7A, although the comparisons between male cones and leaves from different timepoint were not the focus of the manuscript, so that we could provide the total number of DEGs identified.
Comment 10---Fig 7A and text - Please clarify which of two members in a comparison corresponds to "up" or "down".
Response 10: when A is compared to B (A vs B), the reported changes are in A. We have seen this as a common practice. However, if the editor and reviewer persist, we can add additional information in the method.
Comment 11---Tables 1-2 and Figs 8-10 - How many genes that expressed in L1 and L2 are included in the 701 DEGs between M2 and M1. What about for the other three comparisons? I suggest the authors should remove those genes from the pathway analyses.
Response 11: The focus of the study was the differences among male cones at different stages. If these differences also exit among leaves at the same timepoint, it does not change the conclusion of the significance of these DEGs on male cone development, because we did not talk about the exclusiveness in male cone. Therefore, we would argue that it is not unreasonable not to remove DEGs that may be found in leaves as well.
Reviewer 2 Report
Logical errors were corrected:
Verse 55: The correct units of measurement were used.
The size of the features was made more precise:
Verse 202-204: in accordance with Fig. 1A,
Verse 209-210: in accordance with Fig. 1B.
Irregularities in tables and figures have been corrected:
Fig. 2: Verse 221-222,
Fig. 4: Verse 263-264,
Tab. 1: Verse 367-368.
The bibliography was supplemented with 19 publications.
Summary of the review:
The added text fragments, tables (Tab. 2) and charts (Fig. 10) clarify and expand the cognitive scope of the article.
Author Response
Logical errors were corrected:
Verse 55: The correct units of measurement were used.
The size of the features was made more precise:
Verse 202-204: in accordance with Fig. 1A,
Verse 209-210: in accordance with Fig. 1B.
Irregularities in tables and figures have been corrected:
Fig. 2: Verse 221-222,
Fig. 4: Verse 263-264,
Tab. 1: Verse 367-368.
The bibliography was supplemented with 19 publications.
Summary of the review:
The added text fragments, tables (Tab. 2) and charts (Fig. 10) clarify and expand the cognitive scope of the article.
Response: Thank you.
English language and style are fine/minor spell check required
Response: We have corrected some misspellings.
Reviewer 3 Report
The new version of manuscript is well presented and structured, and all the experiments have been carried out correctly and the data analyzed and interpreted as expected. Considering these premises, I recommend the paper for publication.
If possible write in italic the species name reported in Fig.5.
Author Response
Comments and Suggestions for Authors
The new version of manuscript is well presented and structured, and all the experiments have been carried out correctly and the data analyzed and interpreted as expected. Considering these premises, I recommend the paper for publication.
Comment 1---If possible write in italic the species name reported in Fig.5.
Response 1: The suggestion is now included in Fig. 5.
Round 2
Reviewer 1 Report
I would like to ask the authors to answer my questions more precisely. In the response letter, please indicate the line numbers corresponding to the revised part.
1) l. 169 - Did you collect tissue samples for RNA-seq from identical individuals throughout the experiment?
2) l. 155 - "three biological replicates" - Do you mean 'three tissue samples from a tree' or 'one tissue sample from each of three plants'? How about for the RNA-seq sampling?
3) "Response 9: we included all the possible comparisons in Fig. 7A, although the comparisons between male cones and leaves from different timepoint were not the focus of the manuscript, so that we could provide the total number of DEGs identified." - If the authors do not focus on DEGs between male cones and leaves from different timepoint, remove the meaningless data from Fig 7A to avoid confusion.
4) "Response 10: when A is compared to B (A vs B), the reported changes are in A. We have seen this as a common practice. However, if the editor and reviewer persist, we can add additional information in the method." - This information should be described in the manuscript.
Author Response
Reply to the editor and the reviewers
Manuscript No.: forests-1435999
Title: Morpho-histology, endogenous hormone dynamics, and transcriptome profiling
in Dacrydium pectinatum during male cone development
Authors: Wenju Lu , Enbo Wang , Weijuan Zhou , Yifan Li , Zhaoji Li , Xiqiang Song ,
Jian Wang , Mingxun Ren , Donghua Yang , Shaojie Huo , Ying Zhao * ,
Haiying Liang*
Dear editor,
First of all, we thank you for your decision and the reviewers for their valuable comments and suggestions. Each of these comments is not only of great value to our revision in this manuscript but also helpful for us to further our understanding in the studying field. We have revised our manuscript point-by-point after carefully considering the reviewers’ comments. Here, we list below our responses to the comments/suggestions.
Please contact me if there are any other questions. Thank you very much for your help.
We are looking forward to hearing from you soon.
Sincerely yours,
Ying Zhao
----------------------------------------------------------------------------------------------------------------------
Review Report Referee 1
Comments and Suggestions for Authors
I would like to ask the authors to answer my questions more precisely. In the response letter, please indicate the line numbers corresponding to the revised part.
Comment 1--- L. 169 - Did you collect tissue samples for RNA-seq from identical individuals throughout the experiment?
Response 1: yes.
Comment 2--- L. 155 - "three biological replicates" - Do you mean 'three tissue samples from a tree' or 'one tissue sample from each of three plants'? How about for the RNA-seq sampling?
Response 2: For hormone analysis, samples were collected from four trees. After grounding, equal amounts (weight) from individual tree were pooled. We have deleted the sentence about biological replicates. Instead, we said “For each time point, three samples were separately prepared and analyzed.”, considering that they were pooled samples (L145)
For RNAseq, it was 'three tissue samples from a tree' . To make this clearer, we have replaced “There were three biological replicates per time point and tissue type (male bud or leaf).” with “Samples from each tree represented one replicate. Therefore, there were three biological replicates per time point and tissue type (male bud or leaf)” (L155-157).
Comment 3--- "Response 9: we included all the possible comparisons in Fig. 7A, although the comparisons between male cones and leaves from different timepoint were not the focus of the manuscript, so that we could provide the total number of DEGs identified." - If the authors do not focus on DEGs between male cones and leaves from different timepoint, remove the meaningless data from Fig 7A to avoid confusion.
Response 3: Thanks for your suggestion. We have completed the modification of figure 7A (now 6A in the revised manuscript), and changed the relevant results (L. 327~330).
Comment 4--- "Response 10: when A is compared to B (A vs B), the reported changes are in A. We have seen this as a common practice. However, if the editor and reviewer persist, we can add additional information in the method." - This information should be described in the manuscript.
Response 4: Thank you for pointing out the problem. We have added additional information to the method (L. 189-190).
This manuscript is a resubmission of an earlier submission. The following is a list of the peer review reports and author responses from that submission.
Round 1
Reviewer 1 Report
Lu et al. reported the changes in anatomy, hormone physiology, and gene expression during the development of male cones of Dacrydium pectinatum, an endangered conifer species distributed in Southeast Asia. The study likely provided novel information about the development of male cones of the species, however, the data are descriptive on the whole and it is difficult to find significance in terms of conservation biology. General comments:- The authors did not mention why this study is required to conserve the species. In particular, it is not clear for what purpose the authors investigated the anatomy and gene expression during the development of male cones. Although it is mentioned that knowledge about hormones could be useful to promote reproduction (L69–71), no specific suggestion on the conservation in the field was made based on the results in this study. Please provide more information about the current conservation programs (e.g., L60) and explain how the findings in this study is useful to improve that in Discussion.
- The experimental design is unclear as a whole. Which individuals were used for each analysis (e.g., L93, L130, and L140)? Were the samples identical through the anatomical, hormone, and gene expression analyses?
- I did not understand the proposes for statistical analyses of DE genes (L289–337). The authors just compared gene expression between every tissue and stage, providing no biological significance. Which categories of genes are important?
- L17 "To better understand the reproductive process in D. pectinatum" — Please explicitly mention the scientific purposes of the study.
- L25 "GA, IAA, ABA, CTK and JA" — Spell out the abbreviations.
- L93 How did the authors confirm the ages of trees?
- L93 How many trees were used for the measurements? Were the sample trees identical throughout the experimental period?
- L112 Please show the concrete developmental stages.
- L186 "Male cones were mainly found in the well sunlit parts of the outer crown, and 186 there were approximately 45 male cones in a mature D. pectinatum tree (data not shown)." — Unless data and methodology is not provided, this statement is not scientific. Remove the sentence.
- Figure 2 — Ph. and Pg. is not indicated in figures.
- L242 and many other sentences — "first transcriptome" seems an unnecessary overstatement.
- L249 "70,867 unigenes" — The number of unigenes was too many. Probably, the "unigenes" include redundant sequences. The authors should conduct a treatment to remove or cluster redundant sequences (using CD-HIT, for example) before annotation.
- L263–273 and Figure 6 — Annotation about the whole transcriptome data seems meaningless.
- L274 Why did the authors suddenly search SSR sequences, which is not mentioned in Introduction and Method. It is likely SSR markers have been already reported in another study [5].
Author Response
Responses to referee 1:
Comment 1---The authors did not mention why this study is required to conserve the species. In particular, it is not clear for what purpose the authors investigated the anatomy and gene expression during the development of male cones. Although it is mentioned that knowledge about hormones could be useful to promote reproduction (L69–71), no specific suggestion on the conservation in the field was made based on the results in this study. Please provide more information about the current conservation programs (e.g., L60) and explain how the findings in this study is useful to improve that in Discussion.
Response 1: In Introduction of the revised manuscript, we did not add more information about the conservation program because it is not the focus of the manuscript. Instead, we elaborate more in Discussion to show the potential applications derived from the knowledge and resources generated in our study.
Comment 2---The experimental design is unclear as a whole. Which individuals were used for each analysis (e.g., L93, L130, and L140)? Were the samples identical through the anatomical, hormone, and gene expression analyses?
Response 2: All materials in this study were from four mature trees(L100). More information has been provided in the revised manuscript.
Comment 3---I did not understand the proposes for statistical analyses of DE genes (L289–337). The authors just compared gene expression between every tissue and stage, providing no biological significance. Which categories of genes are important?
Response 3: Differential gene expression analysis is a common approach to quickly identify genes that can have crucial roles in a biological process. One disadvantage about this approach is the large number of DE genes. Other associated analyses such as enrichment help narrow the list of candidate genes and pathways. After the general characterization of the DE unigenes, we focused on phytohormone signaling, in addition to the floral timing and organ identity genes, considering the significance of phytohormones in development. The function of these candidate genes needs to be validated experimentally.
Comment 4---L17 "To better understand the reproductive process in D. pectinatum" — Please explicitly mention the scientific purposes of the study.
Response 4: We have specified male cone development in this sentence(L18-L20).
Comment 5---L25 "GA, IAA, ABA, CTK and JA" — Spell out the abbreviations.
Response 5: These have been spelled out as suggested(L24-L26)
Comment 6---L93 How did the authors confirm the ages of trees?
Response 6: We have mentioned in the revised manuscript that the tree age was based on the available records maintained by the state bureau(L101).
Comment 7---L93 How many trees were used for the measurements? Were the sample trees identical throughout the experimental period?
Response 7: Materials for morphological observations were from four trees. One of these trees were used for both endogenous hormone analysis and RNA sequencing. More information has been provided in the revised manuscript(L100-101).
Comment 8---L112 Please show the concrete developmental stages.
Response 8: we have added the collection time period: “from April 2019 to March 2020”(L121-L122).
Comment 9---L186 "Male cones were mainly found in the well sunlit parts of the outer crown, and 186 there were approximately 45 male cones in a mature D. pectinatum tree (data not shown)." — Unless data and methodology is not provided, this statement is not scientific. Remove the sentence.
Response 9: we have deleted this part in the revised manuscript.
Comment 10---Figure 2 — Ph. and Pg. is not indicated in figures.
Response 10: “Ph: phylloclade; Pg: pollen grain.” has been removed in the Figure 2 legend.
Comment 11---L242 and many other sentences — "first transcriptome" seems an unnecessary overstatement.
Response 11: This is the first published transcriptome for the genus, so we want to indicate this information. We have deleted some “first”, changed some into “only” or “initial”, and kept a couple “first”.
Comment 12---L249 "70,867 unigenes" — The number of unigenes was too many. Probably, the "unigenes" include redundant sequences. The authors should conduct a treatment to remove or cluster redundant sequences (using CD-HIT, for example) before annotation.
Response 12: While the genome size of D. pectinatum has not been reported, conifers have large genomes (ranging from 13 to Gb). We suspect that the unigenes contain a large number of isotigs. In the three publications below, the reported number of unigenes was 148,867, 85,305, and 71,669, respectively, in a coniferous species. So the number we obtained does not seem too many.
https://www.ncbi.nlm.nih.gov/pmc/articles/PMC4755536/
https://www.mdpi.com/2073-4425/8/12/393/htm
https://www.nature.com/articles/s41598-019-46696-6
Comment 13---L263–273 and Figure 6 — Annotation about the whole transcriptome data seems meaningless.
Response 13: Figure 6 is a common way to show the overall annotation of a transcriptome. Since this is the first transcriptome being reported for the genus, it would be nice to include such a figure. However, if it has to be left out, we can move it to the supplementary file.
Comment 14---L274 Why did the authors suddenly search SSR sequences, which is not mentioned in Introduction and Method. It is likely SSR markers have been already reported in another study [5].
Response 14: SSR sequences were mentioned because 1) they are useful in diversity and conservation studies, as well as mapping; 2) they provide information about the complexity of the species’ transcriptome; 3) the SSRs reported in [5] were from genomic DNA, and only 15 SSRs were reported. What we report in the manuscript is large-scale.

Reviewer 2 Report
Title: The title of the article is accurate and directly relates to the purpose of the research.
Abstract: The abstract gives a good overview of the work.
Keywords: Keywords are specific to article.
Introduction: The state of the research is reviewed and key publications cited. This study aims to the development of male reproductive structure in Dacrydium pectinatum with a combined approach of anatomy, hormone dynamics, and gene expression.
Verse 51: is "square meter" should be "cubic meter"
Materials and Methods: The empirical material is sufficient in terms of quantity and quality. Appropriate research methods and proper statistical analysis of the data were used.
Results:
Verse 182: value: „diameter of 2.7 mm” – please compare Fig. 1B
Discussion: The discussion is interesting in relation to the publications included in the bibliography. However, please complete the bibliography to references 32-41.
Conclusions: The conclusions are concise and suggest continuation of the research.
Tables and Figures: Developed clearly and visually, please correct the irregularities:
Fig. 2: “Ph: phylloclade; Pg: pollen grain” – please marked on the pictures;
Fig. 4: “The error bars represent standard deviation” – please presented in the figures;
Verse 266: is "Fig. 5" should be "Fig. 6";
Tab. 1: ABA, Total DE unugenes, is "3" should be "4".
References: References include key publications. please complete the bibliography to references 32-41 and missing:
National tropical rain forest protection plan (2016–2020). China Forestry Bureau, 2016.
http://www.forestry.gov.cn/upload?le/main/2016-10/?le/2016-10-10-b4be844ff87944d09ea638695e70c8ac.pdf (in Chinese)
Details in the attached manuscript.
Summary of the review:
The work is important from a scientific and practical perspective for the sake of the population of species Dacrydium pectinatum has been significantly reduced and natural regeneration is poor.
The article fully exhausts the presented issue.

Author Response
Comment 1---Verse 51: is "square meter" should be "cubic meter"
Response 1: “square meter” is replaced by “cubic meter”(L55).
Comment 2---Verse 182: value: „diameter of 2.7 mm” – please compare Fig. 1B
Response 2: The errors have been corrected, “2.7 mm” is replaced by “2.16 mm”, “8.8 mm” is replaced by “8.36 mm”, and “a diameter and a length of 0.5 mm” is replaced by “a diameter of 0.37 mm and a length of 0.32 mm”(Fig.1B)
Comment 3---Discussion: The discussion is interesting in relation to the publications included in the bibliography. However, please complete the bibliography to references 32-41.
Response 3: The missing 32-41 references have been supplemented.
Comment 4---Conclusions: The conclusions are concise and suggest continuation of the research.
Response 4: Thanks.
Comment 5---Fig. 2: “Ph: phylloclade; Pg: pollen grain” – please marked on the pictures;
Response 5: This correction has been made, “Ph: phylloclade; Pg: pollen grain.” has been removed.
Comment 6---Fig. 4: “The error bars represent standard deviation”
– please presented in the figures;
Response 6: Replotted Figure 4, panel A, and error bars were added.
Comment 7---Verse 266: is "Fig. 5" should be "Fig. 6";
Response 7: The errors have been corrected, “Fig. 5” is replaced by “Fig. 6”.
Comment 8---Tab. 1: ABA, Total DE unugenes, is "3" should be "4".
Response 8: The errors have been corrected, “3” is replaced by “4”.
Comment 9---References: References include key publications. please complete the bibliography to references 32-41 and missing.
National tropical rain forest protection plan (2016–2020). China Forestry Bureau, 2016.
http://www.forestry.gov.cn/upload?le/main/2016-10/?le/2016-10-10-b4be844ff87944d09ea638695e70c8ac.pdf (in Chinese)
Details in the attached manuscript.
Response 9: The missing 32-41 references have been supplemented.

Reviewer 3 Report
The manuscript is fragmented and not fluent.The authors should re-construct the experimental design to complete experiments for the manuscript. For example use the RNA sequencing analyses to study the expression of genes linked to endogenous hormones.
Author Response
Comment 1---The manuscript is fragmented and not fluent. The authors should re-construct the experimental design to complete experiments for the manuscript. For example use the RNA sequencing analyses to study the expression of genes linked to endogenous hormones.
Response 1: The manuscript includes three main parts: 1) morphology of the male bud development; 2) endogenous hormone dynamics during male bud development; 3) gene expression during male bud development. The authors feel that all these three parts can be tied together because they represent the morphological, physiological, and molecular aspects of male bud development. In the RNA sequencing analysis, we presented results on genes that are related to hormone signal transduction pathway. This is a term used in annotation to include genes that are endogenous hormones. We have revised the manuscript according to the specific comments from other reviewers with track marks. Hope the revision sufficiently address the matter of the manuscript being “fragmented and not fluent”.

Round 2
Reviewer 1 Report
The authors did not understand my comments and revised the manuscript. Therefore, I cannot recognize the significance of the study. As I mentioned previously, the manuscript lacks the most critical statement on how the study is required and useful. The methodology of statistical analyses of gene expression is not clear and any significant results were presented. I am afraid I cannot recommend publication of the current version of manuscript.
Reviewer 3 Report
Despite the revisions made by the authors, the manuscript does not show a significant improvement.
For this reason, I believe that the article cannot be accepted in the present form.